# Polluted white dwarfs reveal exotic mantle rock types on exoplanets in our solar neighborhood

Keith D. Putirka [1,3✉] & Siyi Xu[2,3]

Prior studies have hypothesized that some polluted white dwarfs record continent-like granitic crust—which is abundant on Earth and perhaps uniquely indicative of plate tectonics. But these inferences derive from only a few elements, none of which define rock type. We thus present the first estimates of rock types on exoplanets that once orbited polluted white dwarfs—stars whose atmospheric compositions record the infall of formerly orbiting planetary objects—examining cases where Mg, Si, Ca and Fe are measured with precision. We find no evidence for continental crust, or other crust types, even after correcting for core formation. However, the silicate mantles of such exoplanets are discernable: one case is Earth like, but most are exotic in composition and mineralogy. Because these exoplanets exceed the compositional spread of >4,000 nearby main sequence stars, their unique silicate compositions are unlikely to reflect variations in parent star compositions. Instead, polluted white dwarfs reveal greater planetary variety in our solar neighborhood than currently appreciated, with consequently unique planetary accretion and differentiation paths that have no direct counterparts in our Solar System. These require new rock classification schemes, for quartz + orthopyroxene and periclase + olivine assemblages, which are proposed here.

[1] Department of Earth and Environmental Sciences, California State University, 2576 E. San Ramon Ave, MS/ST 24, Fresno, CA 93740, USA. [2] Gemini Observatory/NSF's NOIR Lab, #314, 670N. A'ohoku Place, Hilo, HI 96720, USA. [3] These authors contributed equally: Keith D. Putirka, Siyi Xu.
✉email: kputirka@csufresno.edu

White dwarfs have received much attention among exoplanet enthusiasts, as more than a quarter accrete rocky material into their photospheres[1,2]. These so-called "polluted white dwarfs" (PWDs) act as "cosmic mass spectrometers"[3] that provide near-direct analyses of exoplanet compositions. White dwarfs are stars that have left the main sequence, having used up all their fuel; the stars first expand to form red giants, and then contract, to a size that is about that of Earth[4]. At this point, planets orbiting these stars may cross the stellar Roche limit and disintegrate, with the resulting debris falling into the stellar atmospheres[3,4]. Most white dwarfs that have cooled below 25,000 K have atmospheres that consist of pure H or He, as heavier elements sink rapidly to stellar cores at such temperatures[3,4]. When accretion of planetary debris occurs, though, elements heavier than He are detected, giving us our most direct view of exoplanet compositions[3,4]. The pollution sources may consist of entire planets or the broken bits of planets like our asteroid belt[3,4]. But dynamic modeling[5] indicates that metallic cores might be more resistant to tidal forces, so silicate materials (mantle + crust) might be concentrated in pollution sources—which can magnify our view of mantle and crust compositions.

Early studies of PWDs indicate that pollution sources are quite likely dominated by rocky objects, much like our inner planets[4,6–9]. Astronomers often use the term "Earth-like" for such objects to distinguish these from gas giants. But as we will show, PWDs allow for added precision: Mercury, Earth, Moon, and Mars are all "Earth-like" in astronomical terms, but vastly different geologically. We thus reserve the term "Earth-like" for planets that are more similar to Earth than they are to Mars, Mercury, or the Moon, etc., and recommend modifiers such as "Mars-like" or "Mercury-like", etc., as occasion demands.

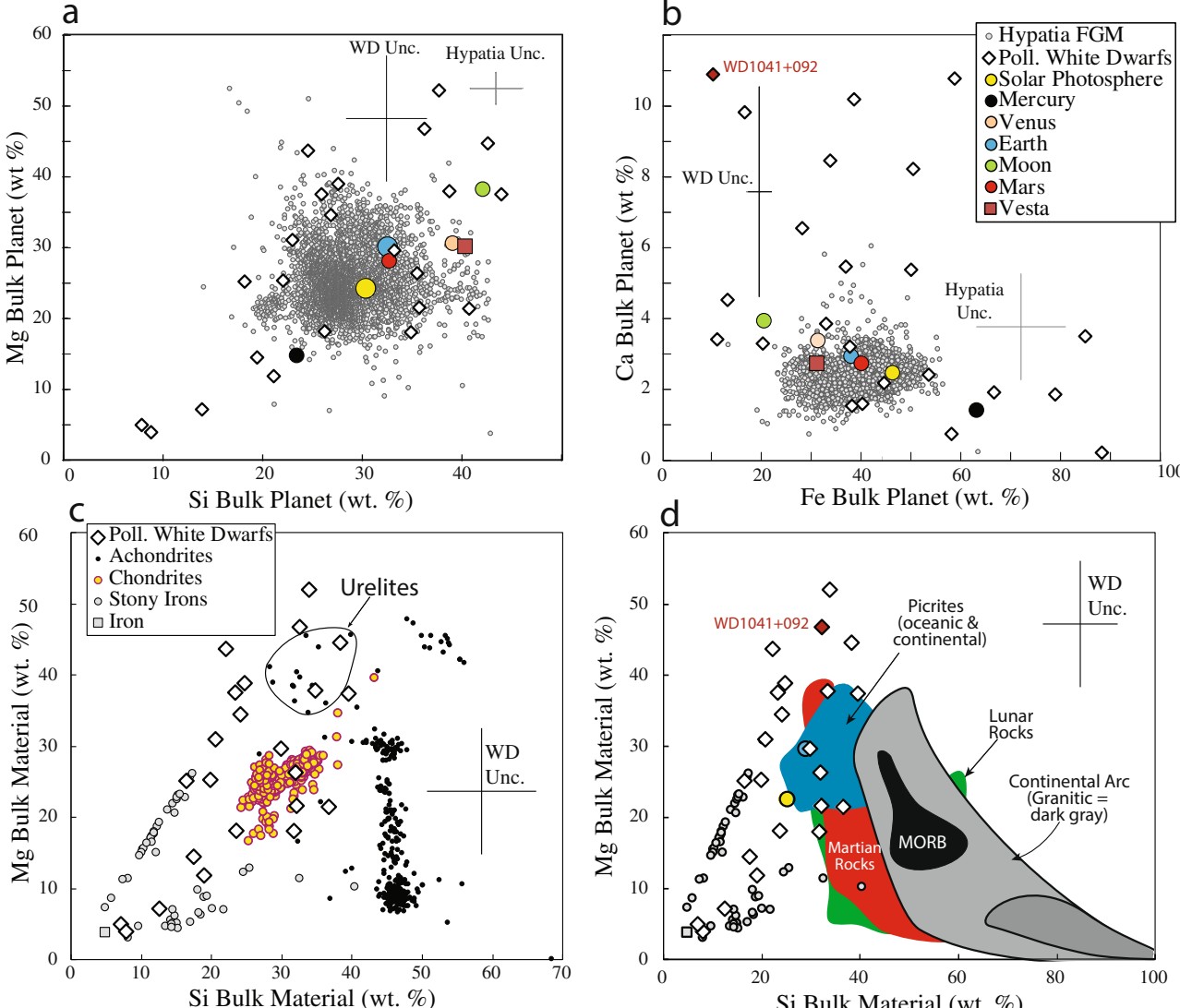

**Fig. 1 A comparison of bulk planet (core + crust + mantle) compositions.** Bulk compositions of polluted white dwarfs (PWDs) are compared to the bulk planets Earth[21], Moon[23], and Mars[26] and FGKM stars of the Hypatia Catalog[17] (**a**, **b**), and various meteorite types[19] (**c**), as well as rocks from Earth[31], Moon[32], and Mars[30], and iron and stony iron meteorites[19] (**d**). **c** also shows the field for the subclass of achondrites known as urelites, which have an unknown parentage. Mg + Si + Ca + Fe are normalized to equal 100%. Vertical and horizontal lines labeled "WD Unc." show the propagated average uncertainties of PWD compositions. **a**, **b** show that PWDs exhibit a much wider range of compositions than that found among FGKM stars. **c**, **d** show that PWDs overlap only imperfectly with meteorites from our Solar System, and almost not at all with rocks from Earth, Mars, or Moon. WD1041 + 092 has the highest Ca among our PWDs, but as can be seen in **d**, it cannot be a candidate for continental crust as it is far removed from granitic rocks that characterize such crust types. MORB Mid-Ocean Ridge Basalt.

Regarding such precision, recent studies assert that continental crust exists on a number of PWDs[10–12]. In one study[11], granitic crust is identified on 27 of 29 PWDs, with granitic mass fractions ranging to 75%. If valid, these would be spectacular finds, as continental crust is a defining characteristic of Earth. Such our only means of identifying exoplanetary plate tectonics, or global water cycles, as continental crust seems necessarily linked to these[13,14]. However, identifications of continental crust are based only on nominally high abundances of Ca and Al[10,11], or ratios of these with Li or other alkali metals[12], usually plotted on a log scale. But none of these elements define rock type, and elevated Li/Ca may be more reflective of galactic-scale chemical evolution than rock compositions[15]. Moreover, none of these studies account for Si, let alone simultaneously examine the sum: Si + Mg + Fe + Ca, which account for >90% of anhydrous cations on nearly all rocky bodies[16,17]. The inclusion of Si is especially critical as it is a hallmark of continental crust, where $SiO_2$ contents average 60 wt% and range to >75%[18]. Our data include 13 PWDs where high granitic crust fractions (30–75%) have been proposed[11], where we identify rock types and test claims of exoplanetary crust compositions.

Here we thus examine 23 PWDs where Ca, Si, Mg, and Fe are measured with precision (see "Methods"). Because PWDs might reflect assimilation of entire planets (mantle + crust + core), we compare their bulk compositions to the bulk compositions of the inner planets of our Solar System, taking estimates of their silicate compositions[19–27] and adding back their metallic cores[28]. Because pollution might be dominated by silicate fractions[5], we also calculate bulk silicate planet (BSP) compositions that account for the removal of Earth-like core fractions[17] from the bulk compositions, and we compare these (as well as bulk PWDs) to meteorites[29], and rocks from Mars[30], Earth[31], and Moon[32], as well as the putative silicate fractions of exoplanets calculated from main sequence star compositions in our galactic neighborhood[16,17].

## Results

We find that our 23 PWDs exhibit compositional ranges that exceed that of the inner planets and the more than 4000 rocky

exoplanet compositions inferred from main sequence stars (Fig. 1a, b). Meteorites capture much of the absolute compositional range of PWDs and a few fall close to chondrites or stony irons (Fig. 1c). However, with their higher Si contents, achondrites[29] and crustal rocks from Mars[30], Earth[31], and Moon[32], all provide poor matches for PWD bulk compositions. Some bulk PWDs overlap in both Mg and Si with a subclass of achondrites called "urelites" (whose origin and parent body are unknown; Fig. 1c), but urelites have much lower Ca[29] than PWDs. High-Ca PWDs, though, overlap with respect to all four elements with a small subset of continental flood basalts[30] (Fig. 1d).

The high Fe in PWDs (Fig. 1b) indicates that some could be assimilating not just silicates but also metal, possibly from differentiated cores. Silica contents are increased if we consider Fe subtraction after core formation and so we calculate "bulk silicate planet" (BSP = mantle + crust) compositions for PWDs (Fig. 2), assuming Earth-like partitioning of Fe between silicate and metal reservoirs (where the fraction of Fe in the mantle relative to the bulk planet, noted as $\alpha_{Fe}$, is 0.27)[16], and compare these to the bulk silicate compositions of Mercury[19], Earth[21], and Mars[25,26], as well as to martian[30], terrestrial[31], and lunar[32] crustal rock compositions (Fig. 2). There is no unique answer to the amount of metallic Fe sequestered into a core, so the precise calculated Fe contents in our BSPs are of little consequence. But even with the ensuing increases in Si, Mg, and Ca, PWDs have $SiO_2$ that is too low and MgO that is too high for any to represent crustal rock types at any significant fraction (Fig. 2). New PWD models show that Mg is often under-estimated, particularly around cool PWDs[33,34]. However, ultramafic mantle rocks from Earth, such as peridotites[21] and pyroxenites[31], are characterized by low $SiO_2$ and high MgO and can explain all but those PWDs that simultaneously range to the lowest $SiO_2$ and highest MgO contents.

## Discussion

Our results verify that PWDs record the accretion of rocky exoplanets, but they also show that those exoplanets associated with PWDs have compositions that are exotic to our Solar System

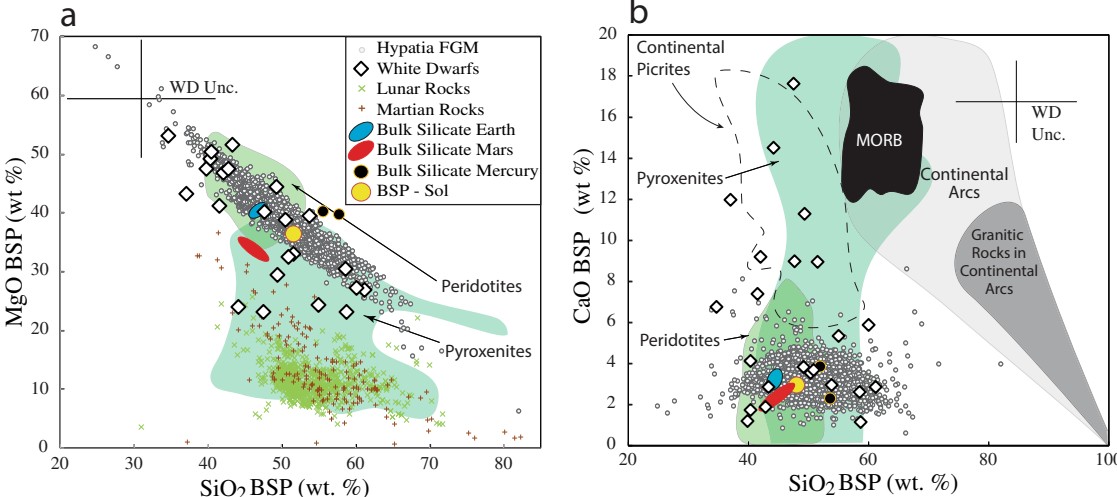

**Fig. 2 A comparison of planetary bulk silicate (mantle + crust) compositions.** Bulk silicate planet (BSP) compositions of polluted white dwarfs (PWDs) are compared to bulk silicate compositions of the inner planets[20–22, 24–27] and the BSPs inferred from Hypatia stars[17], as well as **a** mantle rocks from Earth[21] (peridotites and pyroxenites) and **b** crustal rocks from Earth[31], Moon[32], and Mars. MgO + $SiO_2$ + CaO + FeO are normalized to 100%. BSPs are PWDs when metallic cores are removed from the bulk PWDs of Fig. 1, to allow comparison to silicate compositions of the inner planets. "WD Unc." indicates the propagated average uncertainty of PWD compositions. **b** shows that Earth's continental rocks are a poor match for PWD silicate fractions. **a** and **b** show that rocks from Earth's mantle are a good match for PWDs. But only one PWD matches bulk silicate Earth; most PWDs match rock types that are not dominant on any inner planet. Sol Solar, MORB Mid-Ocean Ridge Basalt.

—sufficiently so to require new rock classification schemes to describe their mineral assemblages (Fig. 3 and Table 1). However, unlike prior studies[11] we find no evidence of continental crust, or sure signs of any high-fraction crustal materials. Some high-Ca PWDs are not inconsistent with their pollution sources being similar to certain Ca-rich Martian meteorites, or rare Ca-rich volcanic rocks erupted in continental flood basalt provinces on Earth (Table 1). However, these same PWDs (e.g., WD1041 + 092, which has the highest Ca in our dataset; Fig. 1b) also have high MgO and low $SiO_2$ (Fig. 1d)—hallmarks of mantle rock types, such as peridotite and pyroxenite. We thus conclude that PWDs record mantle, not crust compositions. This is perhaps not a surprise given that ultramafic mantle rocks are precisely the class of materials we would expect to dominate the silicate fractions of rocky exoplanets: the lunar crust is no more than 10% of the Moon's total mass, while on Earth, the oceanic and continental crusts combine to comprise <0.5% of Earth's total mass and ≈0.7% of its silicate fraction[35].

We are not the first to raise concerns about elevated Ca. Astronomers tend to focus on ratios of Ca to other elements, and hypothesize that high Ca/Mg in some PWDs could reflect preferential sublimation of Mg[7], or that high Ca/Fe involves the loss of Fe during planetary heating, as planets form or when parent

stars undergo a red giant phase[36,37]. Such cases would obviate the need to compare high-Ca PWDs to continental crust or high-Ca mafic rocks from Earth and Mars. In any case, their high Mg and low Si shows the overwhelming likelihood that PWDs record planetary mantles, not crusts. Perhaps most intriguing is that just as the bulk inner planets of our Solar System do not cluster about the Sun (Fig. 1), neither do PWDs precisely mimic the compositions of main sequence stars (Figs. 1 and 2). Studies of Ca/Fe[36] and Na[37,38] in PWDs similarly reveal a wide variety of parent bodies that pollute PWDs, apparently over a considerable range of orbital radii[37]. All these observations show that accretion and planetary differentiation combine to create a wider array of objects than obtained if planets are merely "chondritic" or solar/stellar in bulk composition.

To evaluate this geologic variety we transform PWD compositions into a so-called "normative" or "standard" mineralogy (see Appendix for details), which approximates equilibration at upper mantle conditions on an Earth-sized planet, of ca. 2.0 GPa and 1350 °C[17]. A standard mineralogy facilitates interplanetary comparisons absent various model-dependencies and assumptions, such as water contents, thermal evolution, and pressure–density relationships, which are all unknown but greatly affect mineralogy. Mineral abundances are first plotted into the classic ultramafic rock ternary diagram[39] (Fig. 3a) of olivine $(Mg,Fe)_2$-$SiO_4$ + orthopyroxene $(Mg,Fe)_2Si_2O_6$ + clinopyroxene $Ca(Mg,Fe)Si_2O_6$; these minerals represent >90% of Earth's mantle and are the basis of rocks called "peridotite" and "pyroxenite". Peridotite has >40% olivine, and is the rock type that is expected to also dominate the mantles Moon, Mars, and Mercury (Fig. 3a). Figure 3a thus provides a test of whether PWD materials can be described using the same kinds of rock types that dominate the inner planets of our Solar System. Those PWDs that fall outside such a ternary diagram (Fig. 3a) do so because one or more minerals that form the apices of the ternary are calculated to have negative abundances. In such cases, the PWDs are then recast using new sets of minerals, which leads to the two new ternary classification diagrams (Fig. 3b, c), which can describe PWDs as positive sets of mineral components. Of our 23 PWDs, 11 fall within or adjacent to the ultramafic rock ternary (Fig. 3a; white diamonds), which also describes the mineralogy of the mantles of Mercury, Earth, Moon, and Mars. The remaining PWDs fall well outside this ternary and are exotic to our Solar System in that they lack either olivine or orthopyroxene (both of which dominate the mantles of the inner planets; Fig. 3a). These exotic PWDs either lack olivine and are saturated in quartz (Fig. 3b) or they lack orthopyroxene and are saturated in periclase (Fig. 3c); note that both periclase and quartz are rare to absent from the upper mantles of the inner planets of our Solar System. We thus propose a new naming convention to describe such mantle rock types: "quartz pyroxenites" have >10% each of orthopyroxene, clinopyroxene, and quartz; "quartz orthopyroxenites" have >10% orthopyroxene and quartz, and <10% clinopyroxene; "periclase dunites" have >10% each of periclase and olivine, and <10% pyroxene; "periclase wehrlites" contain >10% each of periclase, olivine, and clinopyroxene; "periclase clinopyroxenites" have <10% olivine and >10% each of periclase and clinopyroxene (Table 1). Note that despite elevated $SiO_2$, none of the five PWDs in Fig. 3c (white diamonds) would qualify as continental crust as they are enriched in orthopyroxene (Fig. 3b) due to their high MgO. If new models of low-temperature PWDs are valid[33,34], then NLTT43806 might also fall into Fig. 3c. It is perhaps worth emphasizing that while thermodynamic models likely lead to insights regarding the geology of some exoplanets[40,41], no current thermodynamic models can predict crust thickness, plate tectonics, or lower mantle mineralogy for the PWDs of Fig. 3b–c, or perhaps even most in Fig. 3a, as partial

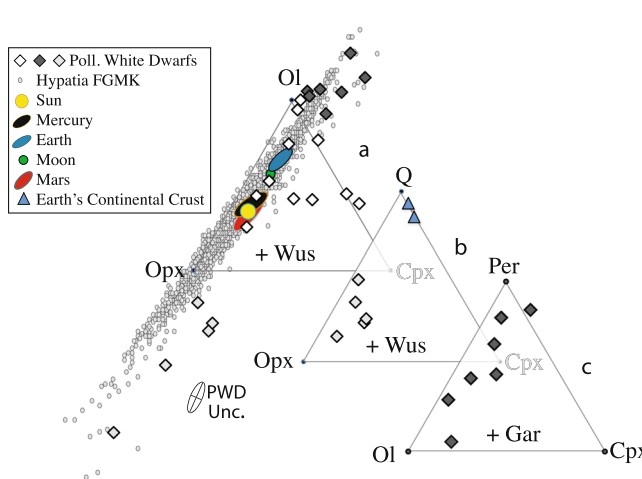

**Fig. 3 Rock Classifications (old and new).** Bulk Silicate Planets (Mg + Si + Ca + Fe) of PWDs are recast as mineral components and plotted in **a** the classic ternary ultramafic rock classification[39], and two new ternary diagrams (**b**, **c**) that can describe PWDs as a set of positive mineral components. "PWD Unc." indicates the propagated average uncertainty in PWD compositions. These ternaries show the relative proportions of the minerals olivine (Ol; $(Mg,Fe)_2SiO_4$), orthopyroxene (Opx; $(Mg,Fe)_2Si_2O_6$), clinopyroxene (Cpx; $Ca(Mg,Fe)Si_2O_6$), periclase (Per; MgO), and quartz (Q; $SiO_2$). The symbols "+ Wus" and "+ Gar" indicate that all rocks within a given ternary respectively contain wüstite (Wus; FeO) or pyrope garnet (Gar; $Mg_3Al_2Si_3O_{12}$). Points falling outside a ternary contain a negative amount of one or more indicated minerals and must be recast using a new set of minerals to obtain positive proportions. Mantle compositions of Earth, Moon Mars, and Mercury fall within (**a**), as do BSPs calculated using FGMK star compositions. Many PWDs fall outside of (**a**), but can be described as positive fractions of either (**b**) Q + Opx + Cpx or (**c**) Per + Ol + Cpx (**c**). Even in **a**, only a few PWDs are mineralogically similar to the inner planets or exoplanets derived from FMGK stars. In **b**, Q-bearing PWD mantles have too much Opx (too much MgO) to represent continental crust, which plots at much higher Q contents. Suggested rock names for the new classification schemes in **b** and **c** are given in Table 1.

**Table 1 Mantle mineral modes and rock types, calculated from bulk silicate planet compositions.**

| White dwarf | Olivine (Mg, Fe)$_2$SiO$_4$ | Clinopyroxene Ca(Mg, Fe)Si$_2$O$_6$ | Orthopyroxene (Mg, Fe)$_2$Si$_2$O$_6$ | Quartz SiO$_2$ | Periclase MgO | Mantle rock type[a] |
|---|---|---|---|---|---|---|
| PG0843 + 517 | | 3.2 | 48.1 | 48.7 | | **Quartz Orthopyroxenite** |
| WD1929 + 011 | 100.6 | 3.3 | −3.9 | | | Dunite |
| WD1536 + 520 | 47.0 | 9.4 | | | 43.6 | **Periclase Dunite** |
| PG1015 + 161 | | 18.2 | 55.8 | 26.0 | | **Quartz Pyroxenite** |
| Ton345 | | 8.8 | 75.6 | 15.6 | | **Quartz Orthopyroxenite** |
| WD1041 + 092 | −3.9 | 20.1 | | | 83.8 | **Periclase Clinopyroxenite** |
| HE0106-3253 | 45.5 | 54.4 | 0.1 | | | Wehrlite |
| GD61 | 74.3 | 10.5 | 15.2 | | | Lherzolite |
| GD40 | 32.5 | 21.6 | | | 45.9 | **Periclase Wehrlite** |
| G241-6 | 24.8 | 11.3 | | | 63.8 | **Periclase Wehrlite** |
| WD1551 + 175 | 42.3 | 29.4 | 28.4 | | | Lherzolite |
| WD2207 + 121 | 52.9 | 11.2 | 35.9 | | | Lherzolite |
| WD1145 + 017 | 25.7 | 13.5 | 60.8 | | | Olivine Websterite |
| WD 1425 + 540 | | 8.2 | 56.3 | 35.5 | | **Quartz Orthopyroxenite** |
| HS2253 + 8023 | 75.6 | 18.0 | | | 6.4 | **Periclase Wehrlite** |
| WDJ0738 + 1835 | 94.5 | 4.9 | 0.6 | | | Dunite |
| WDJ1242 + 5226 | 44.2 | 9.2 | 46.6 | | | Harzburgite or Aubrite |
| G29-38 | | 18.5 | 58.1 | 23.4 | | **Quartz Pyroxenite** |
| WD1232 + 563 | 64.5 | 4.2 | | | 31.2 | **Periclase Dunite** |
| PG1225-079 | 41.6 | 38.9 | 19.5 | | | Lherzolite |
| GD362 | 39.6 | 63.8 | −3.5 | | | Olivine Clinopyroxenite |
| Ross640 | 13.1 | 7.4 | | | 79.5 | **Periclase Dunite** |
| NLTT43806 | 76.6 | 24.6 | −1.2 | | | Wehrlite |
| Sol-BSP[b] | 34.3 | 10.5 | 55.2 | | | Olivine Websterite |

[a]Rock names that are not in bold are for ultramafic rocks from our Solar System[33]; names in bold font are new, proposed names guided by Fig. 3b, c: "quartz pyroxenites" have >10% each of orthopyroxene, clinopyroxene, and quartz; "quartz orthopyroxenites" have >10% orthopyroxene and quartz, and <10% clinopyroxene; "periclase dunites" have >10% each of periclase and olivine, and <10% clinopyroxene; "periclase wehrlites" contain >10% each of periclase, olivine, and clinopyroxene; "periclase clinopyroxenites" have <10% olivine and >10% each of periclase and clinopyroxene.
[b]Sol-BSP is the mineralogy of a rocky planet that has a Solar bulk composition, with an Earth-like amount of Fe partitioned into a metallic core[17].

melting experiments on the relevant compositions have yet to be performed. In addition, while PWDs might record single planets that have been destroyed and assimilated piecemeal[42], the pollution sources might also represent former asteroid belts[5,9], in which case the individual objects of these belts would necessarily be more mineralogically extreme. If current petrologic models[43] may be extrapolated, though, PWDs with quartz-rich mantles (Fig. 3b) might create thicker crusts, while the periclase-saturated mantles (Fig. 3c) could plausibly yield, on a wet planet like Earth, crusts made of serpentinite, which may greatly affect the kinds of life that might evolve on the resulting soils[44]. These mineralogical contrasts should also control plate tectonics[45], although the requisite experiments on rock strength have yet to be carried out.

An interesting result is that, compared to exoplanets inferred to orbit FGMK stars (Fig. 3a), a larger fraction of PWDs fall outside the classic ultramafic rock ternary diagram, and so require our new classification scheme to describe their mantle rock types. This might be an accident, related to our much smaller sampling of PWDs. Another possibility is that FGMK star compositions provide more a view of mean planet composition, and less about absolute lithologic variety in any stellar system. Finally, we cannot rule out the possibility that some PWDs are similar to Earth with respect to crust composition and mineralogy. High K contents identified in LHS 2534 (ref. [12]) are perhaps especially noteworthy, since K is strongly enriched in continental crust. But minor elements, such as K, as well as Li, and Na, do not define rock type and all these elements are fluid mobile and can be enriched in

almost any rock. For example, if high-K PWDs have both high Mg and low Si, they could represent hydrated upper mantle on a water-rich planet, with little direct relevance to crustal compositions. To have a chance at reliable detections of crust compositions (e.g., PWDs with high Si and K combined with low Mg), or plate tectonics via our detections of crust compositions[45,46], we need comprehensive analyses of white dwarfs that include all of what geologists call the "major elements" (Mg, Al, Si, Ca, and Fe) as well as minor elements (Na, K, and Ti) and trace elements that are both highly siderophile (e.g., Ni) and highly lithophile (e.g., U, Ba). Given that Si and Fe vary with galactic radius by orders of magnitude[47], pursuit of these analyses may well show (if corrections can be made for stellar drift within the galaxy) that some parts of the galaxy are more disposed to forming Earth-like planets than others. Exoplanet studies also force us to face still unresolved questions of why Earth is so utterly different from its immediate planetary neighbors, and whether such contrasts are typical or inevitable[48].

## Methods

We focus on 23 PWDs that are located within 200 pc of the Sun, where Mg, Si, Ca, and Fe are detected and uncertainties are reported (Supplementary Table A1). Our tests involve high-Ca PWD of prior studies[9–11], 10 of which are reported to have continental crust fractions, $F_{crust}$, of 30–75%[11] (Table A3). A larger number of elements could be considered, but only at great sacrifice to the total number of PWDs examined ($n = 23$). Table A1 (Supplement) reports star compositions and properties, and published sources. Polluted white dwarfs are compared to star compositions from the Hypatia Catalog[16] which provides compositions of >9000

main sequence (or FGKM-type) stars that fall within 150 pc of the Sun, and where compositions are known with precision. We take a subset of 4200 of these where multiple rock-forming elements are reported[17]. All cation sums for all compositions are renormalized to $Mg + Si + Ca + Fe = 100\%$, or as oxides: $MgO + SiO_2 + CaO + FeO = 100\%$ (for the purposes of comparing Fe in oxidized silicate materials, all Fe is expressed as total FeO, or FeOt). BSP compositions are projected as standard upper mantle mineral components, as employed for a prior study of exoplanets inferred from Hypatia Star compositions[17].

## Data availability

All data used for this study are published in the accompanying Extended Data tables, which are also available from the lead author upon request.

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

## Acknowledgements

K.D.P. was supported by NSF grant #1921182. S.X. was partly supported by the international Gemini Observatory, a program of NSF's NOIRLab, which is managed by the Association of Universities for Research in Astronomy (AURA) under a cooperative agreement with the National Science Foundation on behalf of the Gemini partnership of Argentina, Brazil, Canada, Chile, the Republic of Korea, and the United States of America. K.D.P. thanks John Rarick for inspiring this research path.

## Competing interests

The authors declare no competing interests.
