## [Peer Review File · Nature Communications]

Polluted White Dwarfs Reveal Exotic Mantle Rock Types on Exoplanets in our Solar NeighborhoodReviewers' Comments:

Reviewer #1:

Remarks to the Author:

Key results:

This novel submission introduces three important ideas regarding the chemistry of exoplanetary systems: (1) The widespread trope that bulk metal compositions in white dwarf planetary systems are "Earth-like" is incorrect, (2) We currently don't see evidence of crustal-like exoplanetary material, (3) Exoplanetary chemistry is significantly more varied than solar system chemistry. The submission also, for the first time, identifies a mantle rock type for each of the metal-polluted white dwarfs studied.

Validity:

The validity and interpretation of the data is robust, and the results are convincing. There is one important aspect of the data which I think should be addressed: this submission places a particular focus on the Hollands et al. (2018) study (Reference #11), but does not discuss how their Mg abundances might need to be re-calibrated based on the investigation of Blouin (2020):

<https://ui.adsabs.harvard.edu/abs/2020MNRAS.496.1881B/abstract>

A related relevant study is Turner & Wyatt (2020), who modelled crustal fractions as well as specifically addressed potentially artificially low Mg abundances:

<https://ui.adsabs.harvard.edu/abs/2020MNRAS.491.4672T/abstract>

Significance:

This submission is clearly significant. It introduces novel concepts to the field of white dwarf exoplanetary science, and will help to overturn the "Earth-like" bulk chemical assumption which has become common throughout the field. Also, this submission finally addresses some doubts which have been raised before in a variety of contexts about crustal abundances. Also, the naming of particular mantle rock types in this submission will help to motivate highly detailed geological investigations about white dwarf planetary systems in the future.

Suggested improvements:

(1) As mentioned above, addressing the Mg abundance issue could be important. (2) The claim for not having found evidence for crustal material relies on establishing a clear definition for a crust. This definition could be clarified. Although Line 52 indicates the inclusion of Si is a hallmark of continental crusts, the text does not indicate at what level Si should be present in order to establish a definitive association with a crust. (3) Despite featuring continental crusts, the submission did not cite Jura et al. (2014), which represented a pilot study for the detection of plate tectonics using Ba, Sr and Ca abundances:

<https://ui.adsabs.harvard.edu/abs/2014ApJ...791L..29J/abstract>

(4) The captions of the figures just explain the content, but do not provide any context. A reader, especially from another field, reading those captions would not understand why these figures are important or what they are trying to convey. Also none of the figure captions explain the meaning of the term "WD unc.". (5) I understand that Equations (A1-A4) are meant to include matrices, but the current formatting does not distinguish the columns and rows of the matrices, making the equations nonsensical. (6) The penultimate sentence of the main text appears to assume that stars do not radially migrate within the Galaxy, such that their formation location is similar to their current location. A single phrase or sentence may be added or adjusted to account for this potential movement.

My expertise:

Because I am not a geologist, I cannot evaluate the feasibility of the mantle rock types given in Table

1.

Reviewer #2:

Remarks to the Author:

The authors present a detailed geochemical analysis of polluted white dwarfs and determine the most likely accreted rock types. In particular the authors find no evidence for planetary crusts in the 23 systems they analyse, where silicon provides much stronger constraints than calcium alone. The main issues I find stem from some ignorance of other recent relevant works which are not cited nor discussed/compared against.

Main issues

=====

A large fraction of the submitted manuscript is in the form of a detailed comparison with the Ca/Mg/Fe reprojected of Hollands et al (2018), concluding that prior results do not show evidence for crust accretion. This is presented as one of the main results, however this manuscript is not the first to identify problems with the prior interpretation, yet those other works are not cited here. In particular, Bonsor et al. MNRAS 492, 2683 (2020) dedicate their Section 6 to the anomalously high level Ca/Fe ratio within the Hollands sample, suggesting that heating during planet formation is responsible, and that crust accretion is unlikely to account for such a large fraction of the population. Additionally in Harrison et al, MNRAS 504, 2853 (2021), a reanalysis of DZs in the Hollands sample finds only one object that might be consistent with crust, again finding heating during planetary formation is likely to be responsible for elevated Ca and Ti abundances in some systems. In the more recent work by Hollands et al (2021), those authors note in their opening paragraph that previous studies on crusts are ambiguous, citing Bonsor et al. (2020).

On that note, the more recent, and potentially more convincing claim of continental crust detected by Hollands et al. (2021) is dismissed within a single sentence, where the presented argument is not fully applicable. While all four WDs in that work contain lithium, the crust classification is primarily derived from the extreme potassium abundance in LHS2534, and where lithium was (perhaps erroneously) assumed to be a secondary tracer for crust: potassium is not detected in the other three objects in that work, and they also show a greater degree of lithium enhancement, suggesting those objects are better explained by the Galactic chemical-evolution argument from Kaiser et al. (2021), as stated here. However potassium is not readily burned in stellar interiors as lithium is, and thus cannot be used as a similar tracer of Galactic evolution, ruling out this explanation for LHS2534 (LHS2534 is also probably too young for the Galactic evolution argument to apply). The K/Ca ratio observed for LHS2534 is 1.3 dex greater than for WDJ1644-0449 found by Kaiser et al, indicating very different planetary material accreted by the two objects. As it stands, LHS2534 remains well explained by continental crust accretion, and so arguments to the contrary should demonstrate an alternative mechanism to produce such high potassium abundances.

In the discussion (paragraph starting line 105), a "most intriguing" conclusion is that "accretion and planetary differentiation...combine to create a wider array of objects than would be obtained if planets were merely "chondritic" or solar in bulk composition". The last few years have seen multiple works exploring differentiation as the result of diverse abundances of polluted white dwarfs, yet practically none of those studies are cited in this work. It would be incredibly useful to compare these results with those of these other works, for example:

Swan et al., MNRAS 490, 202 (2019)

Bonsor et al., MNRAS 492, 2683 (2020)

Harrison et al., MNRAS 504, 2853 (2021)

Minor Comments

=====

Line 22: "Solar System" should always be capitalized. Please check throughout.

Line 25: The description of "exoplanet enthusiasts" is a rather odd turn of phrase to see in a scientific manuscript. "Exoplanet researchers" or similar would be better.

Line 32: "When pollution occurs, though, elements heavier than He are detected" -- Given that white dwarfs are classified as polluted from detection of metals, this sentence seems a bit circular. When "accretion of planetary debris occurs" or similar would be better.

Line 53: "the inclusion of Si—a hallmark of continental crust" -- this statement requires a reference (e.g. Rudnick & Gao, 2014), especially seeing as it is considered a key component of the analysis.

Figure 1: The white diamonds do not particularly stand out on these plots compared to other points in the figure. I advise taking some action should be taken to make more prominent, e.g. making the point borders thicker and/or using some other color.

Line 112: "Eleven pWDs fall within this triangle". On inspection of the figure, it seems only six points are contained within the triangle. With the other five white points close to the Ol-Cpx boundary. Do you mean to say "within the uncertainties", or similar? Presumably some of these five points could also appear in (c) as well, or does one of the four elements preclude them from being placed in this panel?

Figure 3: The overlapping ternary diagrams make this figure unnecessarily hard to interpret. Stacking (b) and (c) on the right so that they are clearly separated from each other, and from (a). Is it possible to add the locations of Q and Per to panel (a), so that this figure can more easily be related to (b-c)? Here, the Hypatia catalogue refers to FGKM stars, whereas only FGM are listed in the Fig 1 and 2 legends. Please double check that this is correct.

Line 124: "In addition, while some studies suggest that pWDs record single planets that have been destroyed" Generally

125 and assimilated piecemeal 37

Line 141: "But to have a chance at reliable detections, astronomers must provide more comprehensive analyses of white dwarfs." This is certainly not the result of a lack of trying. The reality is that only a subset of elements are detectable at any given effective temperature e.g. Na is limited to detection in cooler white dwarfs and Si only in warmer white dwarfs. I think this request can be rephrased with a more diplomatic tone.

Methods

Line 158: "A larger number of elements could be considered, but only at great sacrifice to the database size." What do you mean by database? The 23 polluted white dwarfs?

Line 164: "Mg + Si + Ca + Fe = 100", change to "100%" as is the case elsewhere.

Table 1: "Rock names in normal font", do you mean "Roman font"? Sol-BSP should be Solar-BSP.

Table A1: "log of gravitational acceleration", add "in cm/s^2 ". For note (d), I don't know what you mean by (e.g. $Z > 2$) -- Z is used when metals are detected regardless of abundance (see Sion et al, ApJ 269, 253, 1983). For T_{eff} in the table header, the eff should be romanized as in note (e). Similarly

the units for D should also be roman. Given that the units are provided in the header, they do not need repeating in notes (e-f). In scientific English, the name of the Sun is simply "the Sun", not "Sol". The 100 in the log(g) column is confusing -- either remove (it is clear $\text{Si}+\text{Fe}+\text{Mg}+\text{Ca}=100$), or replace with the solar $\log(g)=4.44$.

Table A2: The same point regarding "Sol"/"The Sun" as in Tables 1/A1. It is noteworthy that your analysis finds Ton 345 to be rich in crust+mantle material given that Wilson et al, MNRAS 451, 3237 (2016) report enhanced Fe/Ni, suggesting a stripped mantle.

References

Line 249: In reference 37, "constrains" should be "constraints".

Line 254: In reference 39, "sperpentinite" should be "serpentinite".

Supplementary information

Equations A1--A7: These matrix equations have for some reason been flattened on to single lines, making them almost unreadable. I understand that the authors have used MS Word to typeset the manuscript, however the Word equation editor is more than capable of typesetting matrices. In equations A2/3/5/7 the matrices are incorrectly placed on the right of the multiplication. They might be inverted, but they still operate on the column vectors and not the other way around. More generally the matrix element indices are labels, not variables, and so should not be italicized (except for C_i^j , and similar in the text).

Line 371: "Polluted white dwarf compositions are typically reported as either $[Z/Y]$ " as you note on lines 378-379, the $[Z/H]$ notation is generally understood to mean relative to the Sun's $\log(Z/H)$ value. Therefore I recommend against using the $[Z/Y]$ to mean $\log(Z/Y)$ as this is liable to confuse some readers. Instead just use $\log(Z/Y)$ if possible.

Line 384: Inconsistent spacing around Ca. Note that these logarithmic uncertainties, when translated to linear abundance uncertainties (as in Figures 1-3), will have a different size corresponding to where in the phase-space they are located, e.g. they will shrink towards the bottom left corners in Fig 1.

Line 402/403: Fe, BSP, and BP are labels and so should be in roman font.

Line 404: "Many studies assume that no Fe enters the mantle" please cite some examples of studies that assume this.

Reviewer #3:

Remarks to the Author:

This manuscript is an important response to recent studies that have purported to find evidence for Earth continental crust-like material accreting onto white dwarfs. The important point is made that the few elements used to support that conclusion are not diagnostic of rock type on Earth, much less on exoplanets whose typical mantle and crust mineralogy/modes are unfamiliar to geochemists. This manuscript goes on to show that the polluted white dwarfs exhibit compositions that are much more diverse than those of rocky planets in our solar system or of bulk silicate planets calculated from known star compositions. Perhaps most importantly, the authors present a new classification scheme for exotic mantle rock types. I anticipate that this new terminology will greatly improve

communication with the exoplanet community and will have a long-lasting impact on this developing field. Full comments are in the attached .docx.

Review of: Exotic Exoplanets in Our Galactic Neighborhood

By Putirka and Xu, submitted to *Nature Communications*

Numbers refer to line numbers in the submitted manuscript PDF.

Summary

This manuscript is an important response to recent studies that have purported to find evidence for Earth continental crust-like material accreting onto white dwarfs. The important point is made that the few elements used to support that conclusion are not diagnostic of rock type on Earth, much less on exoplanets whose typical mantle and crust mineralogy/modes are unfamiliar to geochemists. This manuscript goes on to show that the polluted white dwarfs exhibit compositions that are much more diverse than those of rocky planets in our solar system or of bulk silicate planets calculated from known star compositions. Perhaps most importantly, the authors present a new classification scheme for exotic mantle rock types. I anticipate that this new terminology will greatly improve communication with the exoplanet community and will have a long-lasting impact on this developing field.

Overarching Comments

Strengths

Although the main thrust of the manuscript is to show that the polluted white dwarf observations are more likely to represent mantle material rather than rocks from an Earth-like continental crust, I believe that the contribution from this manuscript with the most longevity is the new classification scheme for exoplanet mantle rocks. I strongly recommend that this information be discussed in the main text and positioned more prominently in the abstract.

I think the authors have done a good job of making the astronomy material accessible to a geochemist. The Introduction is a concise primer on the subject, going into the right amount of detail for the subject matter and length of the manuscript.

This was a pleasure to read, and I think the choice of what is shown in the figures well-illustrates the arguments and data presented in the text.

Weaknesses

I have no misgivings about the science underlying this manuscript. The only possible weaknesses I see are related to the figures—their legibility and digestibility. The figures show a great deal of geochemical data, many of which will be unfamiliar to astronomers, and so I suggest copy editing the captions and referring text while keeping an eye on this “novice” segment of the audience. I have made suggests in the line-by-line comments below to this effect and have also addressed some design choices that could use some tweaking. I’m sure other changes could be made as well, to improve the overall reading experience for those who do not have a geochemistry or petrology background.

Line-by-line Comments

Substantive

Abstract: I think the new classification scheme proposed in Table 1 (and hopefully to be included in the text itself!) should be mentioned at the end of the abstract. It is a key outcome of this work and will help exoplanet researchers to better communicate with each other.

Lines 78–79: I suggest softening the language here as I don’t think it is a foregone conclusion that “at least some” PWDs are assimilating metal *cores*, specifically. I suggest something like “The high Fe in PWDs (Fig. 1b) indicates that some pWDs could be assimilating not just silicates but also metal, likely/possibly from differentiated cores”.

Lines 80–81: Please state (or refer to the supplement) the actual value used to mimic the “Earth-like partitioning of Fe between silicate and metal reservoirs”. The manuscript states on lines 83–84 that the precise calculated Fe contents are “of little consequence”, but it is important in the main text to include the value or information about how to obtain it for the sake of transparency/reproducibility.

Figures

Figure 1: Are these bulk planets or bulk silicate planets? The axis labels indicate the former, but the text on line 62 seems to indicate otherwise (see note about the phrasing of this sentence below).

Figure 1d: Why is this PWD labeled? As far as I can tell it is not specifically called out anywhere in the text.

Figure 3: The last sentence of this figure caption refers to the classification scheme used in Table 1, but Table 1 is not referred to here, nor is the new classification scheme discussed in the text. I do suggest removing the “Q- “ from here as it reads as a typo/confusing. Also, why not discuss the classification scheme in the text? This seems like an important outcome of your paper that could guide others in this research area.

Tables

Table 1: I think the text here about the classification scheme should be in the text instead. A reference to the text could then be placed here in the caption.

Editorial

General comments

- Ranges of values should be indicated by an en dash (–) not a hyphen (-). Line 54, for example, should read “30–75%” instead of “30-75%”. Figure panel range references also should use en dashes, not hyphens, as is currently done in the figure captions and text.
 - Sometimes commas are used to indicate two successive panels. Please consistently use either commas or en dashes for this throughout the text and figure captions.
- Small numbers should generally be written out as is done on line 115 “Five” and not shortened to the numeral as is done on line 118 “another 7”.
- There is some inconsistent usage of full terms versus their acronyms. For example, once “polluted white dwarfs” has been shortened to “pWD” (line 26), in my opinion it is better to stick with the acronym. Also, consider changing “pWD” to “PWD” as the former reads as a typo.
- When discussing specific features of a plot in the text, please include a reference to the figure as well as the symbol e.g., line 112 would be more helpful if it read “Eleven PWDs fall within this triangle (Fig. 3a, white diamonds)”.

Line 37: A stray space is in this line between “pollution” and “sources”

Line 40: A typo—this should be “Earth-like” to follow the convention of the rest of the paragraph.

Line 56: This line might be missing an indent

Line 59: A stray comma between “compositions” and “that”

Line 62: The phrasing “This latter nominally” seems to refer to the “putative exoplanets” rather than the “silicate fractions of pWDs” as I think was intended. Please rephrase to clarify.

Line 72: A stray space between “4,000” and “rocky”

Line 76: Typo—“urelites”

Line 79: I suggest to remove the hyphen in “core-formation”.

Line 110: Strongly suggest that you change “triangle” to “ternary” here and throughout the manuscript.

Line 145: Remove hyphen in “un-resolved”

Figures

All figures: Consider moving panel labels outside of the plot boxes, perhaps to above the y axis label, so they are more easily identifiable. This is particularly a problem in Figures 1 and 3. Whatever you choose, please be consistent and place the panel labels in the same place for every figure. Also, the legend text is a bit small and made a bit more difficult to read by how close the symbols are to the text. I suggest the axis labels could be made smaller to increase the plot area and make more room for a more legible legend.

Figure 1a: There are two Mars data points in this panel but I don’t think there is an explanation of how they differ. Also, this is the only panel without tick labels on the x-axis. I understand that they are the same as panel c, but I suggest adding them here anyway.

Figure 1c: The colors (particularly the yellow) in this panel are almost indiscernible. Please change to more saturated colors. Also, the text refers the reader to this panel to see an overlap in PWD data with that of ureilites, but that term does not appear in this panel. Please consider showing ureilitic achondrites with a separate symbol to improve the reading experience.

Figure 1d: The “continental flood basalts” referred to on line 77 are not immediately obvious here. Scientists who are not petrologists or geochemists will read this paper, and so to improve the reading experience, please use the same verbiage in the text as on the figure/legend.

Figure 2: The light gray symbols are just this side of indistinguishable on a white background. I suggest darkening them as they may be completely lost when printed, or when viewed by someone with less acute vision or on a screen with poorer resolution or color reproduction. Also, I suggest improving the utility of the figure caption by changing it thusly, “Bulk silicate planet (BSP) compositions of polluted white dwarfs are compared to bulk silicate compositions of inner planets, (a) mantle rocks from Earth (peridotites and pyroxenites), and (b) crustal rocks from Earth, Moon and Mars. MgO + SiO₂ + CaO + FeO are normalized to 100%.”

Tables

Table 1: I think one of the superscripts here is meant to be a “b” (and then labelled appropriately in the caption). Also, the first instance of “orthopyroxenite” has an unbolded “e” at the end.

Supplements

Supplementary Table A3: The first instance of “orthopyroxenite” has an unbolded “e” at the end

Lines 391–392: I don’t think any uncertainties for the Moon or Vesta are shown on any of the included plots so this line could be edited/deleted.

Line 466: Missing the abbreviation definition for orthopyroxene

Equations 3 and on: There is something odd about the formatting of these equations - maybe a stray indent and one too many returns?

Line 475: Another instance of a hyphen where an en dash is required (I only mention this one because sometimes the supplement is overlooked when making these kinds of changes throughout a document)

Line 476: Typo—should be “Opx” I think?

Lines 476 and 487: “Qtz” and “Per” are used here but were not previously defined with the mineral phases on lines 465–466. “Wüstite” is written out when it should probably be abbreviated for consistency.

Reviewer 1:

“There is one important aspect of the data which I think should be addressed: this submission places a particular focus on the Hollands et al. (2018) study (Reference #11), but does not discuss how their Mg abundances might need to be re-calibrated based on the investigation of Blouin (2020). A related relevant study is Turner & Wyatt (2020), who modelled crustal fractions as well as specifically addressed potentially artificially low Mg abundances:”

We thank the reviewer for making us aware of these studies. The new calibrations are now cited as items 32 and 33. It seems that this model applies to cool WDs (<8,000 K) and we only have one WD that would be affected (NLTT 43806). In any case, where valid, these results would cause MgO contents to be even higher than we show, driving polluted white dwarf compositions further from granitic crust and closer to mantle peridotite compositions – which we now note in the revised draft.

Suggested improvements:

“(1) As mentioned above, addressing the Mg abundance issue could be important.”

We've addressed this, as noted above.

“(2) The claim for not having found evidence for crustal material relies on establishing a clear definition for a crust. This definition could be clarified. Although Line 52 indicates the inclusion of Si is a hallmark of continental crusts, the text does not indicate at what level Si should be present in order to establish a definitive association with a crust.”

A very fair point. We now indicate numerical values for mean continental crust and an expected range. To support the statements, we've moved the original citation #36 to become the new citation #18, with wt. % SiO₂ values listed, providing the average and range for SiO₂ in continental crust.

“(3) Despite featuring continental crusts, the submission did not cite Jura et al. (2014), which represented a pilot study for the detection of plate tectonics using Ba, Sr and Ca abundances: <https://ui.adsabs.harvard.edu/abs/2014ApJ...791L..29J/abstract>”

We've added that citation to the end of the discussion.

“(4) The captions of the figures just explain the content, but do not provide any context. A reader, especially from another field, reading those captions would not understand why these figures are important or what they are trying to convey. Also none of the figure captions explain the meaning of the term ‘WD unc.’”

This is a very fair point and so we've expanded our figure captions, with comments that are directed specifically towards a non-geologic audience (and in each case now define "WD unc.", which represents our propagated uncertainties on white dwarf compositions).

"(5) I understand that Equations (A1-A4) are meant to include matrices, but the current formatting does not distinguish the columns and rows of the matrices, making the equations nonsensical."

Yes, nonsense they are. For some reason, the formats using the equation editor in MS Word do not hold in some versions of our document. For the time being, we've abandoned the MS Word equation editor and have pasted the equations as pdf images. Of course at some later time we'll be more than happy to work with typesetters to submit these equations in any format that works for them.

"(6) The penultimate sentence of the main text appears to assume that stars do not radially migrate within the Galaxy, such that their formation location is similar to their current location. A single phrase or sentence may be added or adjusted to account for this potential movement."

We've added a qualifying clause to this sentence.

Reviewer 2:

"A large fraction of the submitted manuscript is in the form of a detailed comparison with the Ca/Mg/Fe reprojection of Hollands et al (2018), concluding that prior results do not show evidence for crust accretion. This is presented as one of the main results, however this manuscript is not the first to identify problems with the prior interpretation, yet those other works are not cited here. In particular, Bonsor et al. MNRAS 492, 2683 (2020) dedicate their Section 6 to the anomalously high level Ca/Fe ratio within the Hollands sample, suggesting that heating during planet formation is responsible, and that crust accretion is unlikely to account for such a large fraction of the population. Additionally in Harrison et al, MNRAS 504, 2853 (2021), a reanalysis of DZs in the Hollands sample finds only one object that might be consistent with crust, again finding heating during planetary formation is likely to be responsible for elevated Ca and Ti abundances in some systems. In the more recent work by Hollands et al (2021), those authors note in their opening paragraph that previous studies on crusts are ambiguous, citing Bonsor et al. (2020)."

We thank the reviewer for bringing these papers to our attention and noting the relevant details. We now cite both Bonsor et al. (2020) and Harrison et al. (2021). We were aware of some skepticism about the source of high Ca as in our original draft we had cited Xu et al. (2019) for their thoughts that sublimation might affect Ca/Fe ratios. In the new draft, we cite these two papers in this same sentence, adding a clause to note that planetary heating during either during the formation stage or the red giant phase may be another path to increase Ca/Fe ratios

"On that note, the more recent, and potentially more convincing claim of continental crust detected by Hollands et al. (2021) is dismissed within a single sentence, where the presented

argument is not fully applicable. While all four WDs in that work contain lithium, the crust classification is primarily derived from the extreme potassium abundance in LHS2534, and where lithium was (perhaps erroneously) assumed to be a secondary tracer for crust: potassium is not detected in the other three objects in that work, and they also show a greater degree of lithium enhancement, suggesting those objects are better explained by the Galactic chemical-evolution argument from Kaiser et al. (2021), as stated here. However potassium is not readily burned in stellar interiors as lithium is, and thus cannot be used as a similar tracer of Galactic evolution, ruling out this explanation for LHS2534 (LHS2534 is also probably too young for the Galactic evolution argument to apply). The K/Ca ratio observed for LHS2534 is 1.3 dex greater than for WDJ1644-0449 found by Kaiser et al, indicating very different planetary material accreted by the two objects. As it stands, LHS2534 remains well explained by continental crust accretion, and so arguments to the contrary should demonstrate an alternative mechanism to produce such high potassium abundances.”

Possibly. To be sure, observations of high K are intriguing. In our original draft, we closed our discussion noting that “We cannot rule out the possibility that some pWDs are truly similar to Earth with respect to crust...composition and mineralogy...”.

We now add the clause “High K contents identified in LHS 2534¹² might indicate crust compositions. But minor elements, such as K, Li, and Na, do not define rock type and all these elements are fluid mobile—if high-K pWDs have high Mg and low Si, for example, they could instead represent hydrated upper mantle on a water-rich planet, with little direct relevance to crust compositions.”

We do not note this in the paper, but terrestrial peridotites (mantle rocks from Earth) have K/Ca ratios that range to 2.35, which is nearly 1.6 dex greater than the solar value. To be fair to the reviewers, most peridotites have much lower K/Ca (avg. = 0.09), but as we note, K is fluid-mobile. So we’re not confident that high K alone would signify continental crust. However, in combination, high K and Si, combined with low Mg would be a strong indicator of continental-like crust and would be a very exciting find, as we now note later in the same paragraph.

“In the discussion (paragraph starting line 105), a “most intriguing” conclusion is that “accretion and planetary differentiation...combine to create a wider array of objects than would be obtained if planets were merely “chondritic” or solar in bulk composition”. The last few years have seen multiple works exploring differentiation as the result of diverse abundances of polluted white dwarfs, yet practically none of those studies are cited in this work. It would be incredibly useful to compare these results with those of these other works, for example:

Swan et al., MNRAS 490, 202 (2019)

Bonsor et al., MNRAS 492, 2683 (2020)

Harrison et al., MNRAS 504, 2853 (2021)”

We thank the reviewers for pointing out these studies. We agree that these are relevant and important findings and so we now cite and very briefly discuss these in the revised Discussion section, just after the sentence noted in this comment.

Minor Comments

=====

“Line 22: “Solar System” should always be capitalized. Please check throughout.”

Changed as suggested

“Line 25: The description of “exoplanet enthusiasts” is a rather odd turn of phrase to see in a scientific manuscript. “Exoplanet researchers” or similar would be better.”

Somebody has to be “odd”, otherwise the world would be boring

“Line 32: “When pollution occurs, though, elements heavier than He are detected” -- Given that white dwarfs are classified as polluted from detection of metals, this sentence seems a bit circular. When “accretion of planetary debris occurs” or similar would be better.”

Agreed. Thanks for spotting that– changed as suggested.

“Line 53: “the inclusion of Si—a hallmark of continental crust” -- this statement requires a reference (e.g. Rudnick & Gao, 2014), especially seeing as it is considered a key component of the analysis.”

Agreed – changed as suggested (also per the recommendation of R1 and their suggestion to quantify SiO₂ contents)

“Figure 1: The white diamonds do not particularly stand out on these plots compared to other points in the figure. I advise taking some action should be taken to make more prominent, e.g. making the point borders thicker and/or using some other color.”

In each of the figures, we’ve enlarged the white diamonds slightly and greatly thickened the lines that form their borders. We think they stand out more now.

“Line 112: “Eleven pWDs fall within this triangle”. On inspection of the figure, it seems only six points are contained within the triangle. With the other five white points close to the Ol-Cpx boundary. Do you mean to say “within the uncertainties”, or similar? Presumably some of these five points could also appear in (c) as well, or does one of the four elements preclude them from being placed in this panel?”

Yes, good point and correctly inferred by the reviewer. In the revised text, we now write “Eleven PWDs fall within or adjacent to this triangle”

“Figure 3: The overlapping ternary diagrams make this figure unnecessarily hard to interpret. Stacking (b) and (c) on the right so that they are clearly separated from each other, and from (a). Is it possible to add the locations of Q and Per to panel (a), so that this figure can more easily be related to (b-c)?”

Not really, as it would require a 3D or even a 4D figure for the projection (we're projecting 4 components into a 2D plane). And without the overlap, we leave a lot of empty white space. There is also another reason for using these figures: they will be familiar to geologists and are the standard way of portraying mineral-based classification schemes for rocks (including our overlapping triangle display). For astronomers and geologists to talk to one another, a familiarity with these kinds of diagrams is critical.

Here, the Hypatia catalogue refers to FGМК stars, whereas only FGM are listed in the Fig 1 and 2 legends. Please double check that this is correct.”

Thanks for noting these. We now write “FGМК” as we do have some K-type stars in the projections.

“Line 124: ‘In addition, while some studies suggest that pWDs record single planets that have been destroyed’ Generally”

125 and assimilated piecemeal 37”

We're not sure what comment R2 has in mind and so we have made no edits to the sentence that is on lines 207-208 of the revised draft.

Line 141: “But to have a chance at reliable detections, astronomers must provide more comprehensive analyses of white dwarfs.” This is certainly not the result of a lack of trying. The reality is that only a subset of elements are detectable at any given effective temperature e.g. Na is limited to detection in cooler white dwarfs and Si only in warmer white dwarfs. I think this request can be rephrased with a more diplomatic tone.

Agreed. And we actually made the change (deleting “astronomers”) before seeing this comment. The new sentence reads:

“To have a chance at reliable detections of crust compositions (e.g., PWDs with high Si and K combined with low Mg), or plate tectonics⁴⁵⁻⁴⁶, we need comprehensive analyses of white dwarfs that include the “major elements” (Ca, Mg, Al, Si and Fe) and trace elements that are both highly siderophile (e.g., Ni) and highly lithophile (e.g., U, Ba).”

Methods

Line 158: “A larger number of elements could be considered, but only at great sacrifice to the database size.” What do you mean by database? The 23 polluted white dwarfs?

Yes. We now write: “...great sacrifice to the total number of polluted white dwarfs examined (n=23).”

Line 164: “Mg + Si + Ca + Fe = 100”, change to “100%” as is the case elsewhere.

Good point. Changed as suggested.

Table 1: “Rock names in normal font”, do you mean “Roman font”? Sol-BSP should be Solar-BSP.

Agreed – not clear. We now write: “...Rock names that are not in bold italics are...” as we want to contrast these with rock names that are in bold and italicized.

Table A1: “log of gravitational acceleration”, add “in cm/s²”.

Changed as suggested

For note (d), I don’t know what you mean by (e.g. $Z > 2$) -- Z is used when metals are detected regardless of abundance (see Sion et al, ApJ 269, 253, 1983).

Good point. We were using Z as both a classification system for PWDs and as the general symbol for “atomic number”. We now write “(e.g., atomic number > 2)”. This qualification is important because for geologists elements such as Ar, for example, are not metals.

“For Teff in the table header, the eff should be romanized as in note (e). Similarly the units for D should also be roman. Given that the units are provided in the header, they do not need repeating in notes (e-f). In scientific English, the name of the Sun is simply “the Sun”, not “Sol”. The 100 in the log(g) column is confusing -- either remove (it is clear $\text{Si} + \text{Fe} + \text{Mg} + \text{Ca} = 100$), or replace with the solar $\log(g) = 4.44$.”

Thanks for reading the paper in such detail. All of these suggested edits are changed as suggested.

Table A2: The same point regarding “Sol”/”The Sun” as in Tables 1/A1.

Changed as suggested (we searched the entire document so as to replace “Sol” with “Sun”).

It is noteworthy that your analysis finds Ton 345 to be rich in crust+mantle material given that Wilson et al, MNRAS 451, 3237 (2016) report enhanced Fe/Ni, suggesting a stripped mantle.

Interesting. We re-ran the analysis for Ton 345 using the data from Wilson et al., which are definitely higher in Fe than the Jura et al. composition that we used. Unsurprisingly, we thus obtain a larger core fraction: 45% Core + 55% Mantle. But as one can see, we don’t obtain a planet that has been completely stripped of its mantle. And this result is not tied to our particular mass balance approach. Even in a very large parent body, it would be highly unlikely that a metallic core could dissolve the full complements of Mg, Si and O. etc., as reported in their Table 3.

References

Line 249: In reference 37, “constrains” should be “constraints”.

Line 254: In reference 39, “sperpentinite” should be “serpentinite”.

Thank you for your detailed reading and catching both of those errors.

Supplementary information

Equations A1--A7: These matrix equations have for some reason been flattened on to single lines, making them almost unreadable. I understand that the authors have used MS Word to typeset the manuscript, however the Word equation editor is more than capable of typesetting matrices. In equations A2/3/5/7 the matrices are incorrectly placed on the right of the multiplication. They might be inverted, but they still operate on the column vectors and not the other way around. More generally the matrix element indices are labels, not variables, and so should not be italicized (except for C_i^j , and similar in the text).

This was all a mess, as also noted by R1. We did indeed use the MS Word equation editor. But for some reason, the equations do not hold (even in multiple earlier versions that we have saved on our computers). In any case, we've addressed all the issues raised here (deleted italics except for C_i , switched order of row and square matrices on products side, etc.), and to preserve formatting, we've posted the equations as pdf objects.

Line 371: “Polluted white dwarf compositions are typically reported as either $[Z/Y]$ ” as you note on lines 378-379, the $[Z/H]$ notation is generally understood to mean relative to the Sun's $\log(Z/H)$ value. Therefore I recommend against using the $[Z/Y]$ to mean $\log(Z/Y)$ as this is liable to confuse some readers. Instead just use $\log(Z/Y)$ if possible.

Good point. Changed as suggested.

Line 384: Inconsistent spacing around Ca. Note that these logarithmic uncertainties, when translated to linear abundance uncertainties (as in Figures 1-3), will have a different size corresponding to where in the phase-space they are located, e.g. they will shrink towards the bottom left corners in Fig 1.

Agreed. We plotted average uncertainties because if we plotted error bars for each data point, it's more difficult to read the diagrams. We could alternatively plot two error bars, one in the upper right and another in the lower left of these figures.

Line 402/403: Fe, BSP, and BP are labels and so should be in roman font.

Changed as corrected (leaving Fe in italics as it is a variable)

Line 404: “Many studies assume that no Fe enters the mantle” please cite some examples of studies that assume this.

We've deleted this sentence. We were thinking of Howard et al. (2013), Unterborn et al. (2018), Hinkel and Unterborn (2018) and some earlier studies. But even in Unterborn's and Hinkel's

recent papers, they account for mantle $\text{Fe}/\text{Mg} > 0$, and most recent studies from other workers also account for Fe in the mantle. So upon reflection we feel our original sentence is not valid.

Reviewer 3:

Overarching Comments

Strengths Although the main thrust of the manuscript is to show that the polluted white dwarf observations are more likely to represent mantle material rather than rocks from an Earth-like continental crust, I believe that the contribution from this manuscript with the most longevity is the new classification scheme for exoplanet mantle rocks. I strongly recommend that this information be discussed in the main text and positioned more prominently in the abstract.

We now note in the abstract that two new classification schemes are proposed, to describe the mineralogy/rock types of the more exotic exoplanet rock types. We have also re-written the opening paragraph of the Discussion to emphasize this aspect.

I think the authors have done a good job of making the astronomy material accessible to a geochemist. The Introduction is a concise primer on the subject, going into the right amount of detail for the subject matter and length of the manuscript.

This was a pleasure to read, and I think the choice of what is shown in the figures well-illustrates the arguments and data presented in the text.

Weaknesses

I have no misgivings about the science underlying this manuscript. The only possible weaknesses I see are related to the figures—their legibility and digestibility. The figures show a great deal of geochemical data, many of which will be unfamiliar to astronomers, and so I suggest copy editing the captions and referring text while keeping an eye on this “novice” segment of the audience. I have made suggests in the line-by-line comments below to this effect and have also addressed some design choices that could use some tweaking. I’m sure other changes could be made as well, to improve the overall reading experience for those who do not have a geochemistry or petrology background.

We’ve enhanced the figure captions and in the revised captions try to explain some key terms and, pointers on how to read the diagrams – especially the triangular classification diagrams which will probably be the most challenging for astronomers. In addressing one of the key suggestions here, we also have a much expanded discussion of the classification systems (deleting other material to maintain a short article). We hope that we have decreased the effort that astronomers will have to make to understand and communicate with geologists.

Line-by-line Comments

Substantive

Abstract: I think the new classification scheme proposed in Table 1 (and hopefully to be included in the text itself!) should be mentioned at the end of the abstract. It is a key outcome of this work and will help exoplanet researchers to better communicate with each other.

Changed as suggested. The abstract now closes with the following: **“These [exotic exoplanets] require new rock classification schemes, for quartz + orthopyroxene and periclase + olivine mineralogies, which are proposed here.”**

Lines 78–79: I suggest softening the language here as I don’t think it is a foregone conclusion that “at least some” PWDs are assimilating metal cores, specifically. I suggest something like “The high Fe in PWDs (Fig. 1b) indicates that some pWDs could be assimilating not just silicates but also metal, likely/possibly from differentiated cores”.

We’ve adopted the sentence as suggested.

Lines 80–81: Please state (or refer to the supplement) the actual value used to mimic the “Earthlike partitioning of Fe between silicate and metal reservoirs”. The manuscript states on lines 8384 that the precise calculated Fe contents are “of little consequence”, but it is important in the main text to include the value or information about how to obtain it for the sake of transparency/reproducibility.

We’ve added a parenthetical sentence describing the Fe partitioning ratio, and giving its value.

Figures

Figure 1: Are these bulk planets or bulk silicate planets? The axis labels indicate the former, but the text on line 62 seems to indicate otherwise (see note about the phrasing of this sentence below).

Here, we meant “rocky planet” to refer to our inner planets, rather than gas giants. But we can see why this would be unclear, so we’ve deleted “rocky” which we think solves the problem.

Figure 1d: Why is this PWD labeled? As far as I can tell it is not specifically called out anywhere in the text.

Correct. It was called out in an earlier version and then deleted to shorten the paper. We’ve added a note to the figure caption and re-inserted a comment into the discussion. WD1041 is a case of a pWD, having the highest Ca amount our set of pWDs, but with Mg and Si contents that are so high as to be completely inappropriate for continental crust.

Figure 3: The last sentence of this figure caption refers to the classification scheme used in Table 1, but Table 1 is not referred to here, nor is the new classification scheme discussed in the text. I do suggest removing the “Q- “ from here as it reads as a typo/confusing. Also, why not discuss the classification scheme in the text? This seems like an important outcome of your paper that could guide others in this research area.

We've corrected the typo in Fig. 3 and have re-written what is now the fourth paragraph of the Discussion section to better describe our new classification scheme (and in the process, delete some repetitive and less useful text to maintain the current ms. length).

Tables

Table 1: I think the text here about the classification scheme should be in the text instead. A reference to the text could then be placed here in the caption.

Changed as suggested, with a completely re-written 4th paragraph in the Discussion section. (We retain the description of the rock types in the Table 1 caption as we presume that it costs very little with regard to space, and might be convenient to readers examining the table).

Editorial

General comments

- Ranges of values should be indicated by an en dash (–) not a hyphen (-). Line 54, for example, should read “30–75%” instead of “30-75%”. Figure panel range references also should use en dashes, not hyphens, as is currently done in the figure captions and text.

Changed as suggested (we searched the document and found two other instances of the same, which are now fixed).

- o Sometimes commas are used to indicate two successive panels. Please consistently use either commas or en dashes for this throughout the text and figure captions.

Changed to en dashes throughout (if our search correctly caught all instances).

- Small numbers should generally be written out as is done on line 115 “Five” and not shortened to the numeral as is done on line 118 “another 7”.

Changed as suggested (I think we've caught all instances)

- There is some inconsistent usage of full terms versus their acronyms. For example, once “polluted white dwarfs” has been shortened to “pWD” (line 26), in my opinion it is better to stick with the acronym. Also, consider changing “pWD” to “PWD” as the former reads as a typo.

Changed as suggested. We now only spell out “polluted white dwarfs” (now “PWD”) in the first use in the Introduction, and in first use in Figure 1 (but not in other figures).

- When discussing specific features of a plot in the text, please include a reference to the figure as well as the symbol e.g., line 112 would be more helpful if it read “Eleven PWDs fall within this triangle (Fig. 3a, white diamonds)”.

Changed as suggested

Line 37: A stray space is in this line between “pollution” and “sources” Line 40: A typo—this should be “Earth-like” to follow the convention of the rest of the paragraph.

Fixed; changed as suggested

Line 56: This line might be missing an indent

Agreed. Indent added.

Line 59: A stray comma between “compositions” and “that”

The sentence in question is now re-written. We’ve searched the document to be sure that commas do not precede “that”

Line 62: The phrasing “This latter nominally” seems to refer to the “putative exoplanets” rather than the “silicate fractions of pWDs” as I think was intended. Please rephrase to clarify.

Agreed. This sentence is re-written.

Line 72: A stray space between “4,000” and “rocky”

Deleted

Line 76: Typo–“urelites”

Quotes added

Line 79: I suggest to remove the hyphen in “core-formation”.

Agreed. Done.

Line 110: Strongly suggest that you change “triangle” to “ternary” here and throughout the manuscript.

Agreed. Changed as suggested, throughout.

Line 145: Remove hyphen in “un-resolved”

Done

Figures

All figures: Consider moving panel labels outside of the plot boxes, perhaps to above the y axis label, so they are more easily identifiable. This is particularly a problem in Figures 1 and 3.

Whatever you choose, please be consistent and place the panel labels in the same place for every figure. Also, the legend text is a bit small and made a bit more difficult to read by how close the symbols are to the text. I suggest the axis labels could be made smaller to increase the plot area and make more room for a more legible legend.

All changed made as suggested

Figure 1a: There are two Mars data points in this panel but I don't think there is an explanation of how they differ. Also, this is the only panel without tick labels on the x-axis. I understand that they are the same as panel c, but I suggest adding them here anyway.

Tick labels are back in (our original misstate – they were never supposed to be absent). We've also deleted one of the Mars bulk compositions and now realize that the data sources were not noted in the captions – in the revised captions we've added citations for the sources of data for each panel.

Figure 1c: The colors (particularly the yellow) in this panel are almost indiscernible. Please change to more saturated colors. Also, the text refers the reader to this panel to see an overlap in PWD data with that of ureilites, but that term does not appear in this panel. Please consider showing ureilitic achondrites with a separate symbol to improve the reading experience. Figure 1d: The “continental flood basalts” referred to on line 77 are not immediately obvious here. Scientists who are not petrologists or geochemists will read this paper, and so to improve the reading experience, please use the same verbiage in the text as on the figure/legend.

Excellent suggestions and we've followed all of these. We've simplified the display a bit by putting all the chondrites into a single symbol color (now a saturated yellow with a reddish border). And to indicate the urelites, which are a subclass of the achondrites, we've drawn a circle around that subset and add a label (“Urelites”, of course). We also note in the caption and the main text that these are of an unknown parent body/origin.

Figure 2: The light gray symbols are just this side of indistinguishable on a white background. I suggest darkening them as they may be completely lost when printed, or when viewed by someone with less acute vision or on a screen with poorer resolution or color reproduction.

Changed as suggested (on the grayscale border we increased % black from 25% to 75%). The same symbols in Figs. 1 and 3 are also changed in this way, for consistency and readability.

Also, I suggest improving the utility of the figure caption by changing it thusly, “Bulk silicate planet (BSP) compositions of polluted white dwarfs are compared to bulk silicate compositions of inner planets, (a) mantle rocks from Earth (peridotites and pyroxenites), and (b) crustal rocks from Earth, Moon and Mars. MgO + SiO₂ + CaO + FeO are normalized to 100%.”

We've adopted this sentence, and have included additional text as noted by this and other reviewers, to clarify the meanings of each figure.

Tables

Table 1: I think one of the superscripts here is meant to be a “b” (and then labelled appropriately in the caption). Also, the first instance of “orthopyroxenite” has an unbolded “e” at the end.

Fixed as suggested. (Thank you for the very careful reading of the ms.)

Supplements

Supplementary Table A3: The first instance of “orthopyroxenite” has an unbolded “e” at the end

Fixed.

Lines 391–392: I don’t think any uncertainties for the Moon or Vesta are shown on any of the included plots so this line could be edited/deleted.

Correct (and deleted). We had multiple lunar bulk compositions in a prior versions of the figures, but simplified these with averages.

Line 466: Missing the abbreviation definition for orthopyroxene

Fixed

Equations 3 and on: There is something odd about the formatting of these equations - maybe a stray indent and one too many returns?

Yes. Odd. As noted by the other reviewers, we had problems getting the MS Word equation editor to hold on to its formatting. So we have re-pasted the equations as pdf objects.

Line 475: Another instance of a hyphen where an en dash is required (I only mention this one because sometimes the supplement is overlooked when making these kinds of changes throughout a document)

And we did overlook it. Thanks (and corrected).

Line 476: Typo—should be “Opx” I think?

Yes. Thanks.

Lines 476 and 487: “Qtz” and “Per” are used here but were not previously defined with the mineral phases on lines 465–466. “Wüstite” is written out when it should probably be abbreviated for consistency.

Yes. Thanks. Fixed as noted.

Reviewers' Comments:

Reviewer #1:

Remarks to the Author:

I am satisfied with the authors' revisions and now would be happy to see this submission published.

Reviewer #2:

Remarks to the Author:

I thank the authors for either addressing or responding to practically all of my comments. There are some minor issues remaining with the matrix equations in the appendix as well as a new typo, but none of these should not take too long to fix:

* The majority of the matrix indices have been correctly changed to Roman font except for in eqs A1--A2.

* The matrices and vectors are still transposed with respect to convention. I.e. matrices should be operating on column vectors with each matrix row being multiplied element-wise with the column vector. Although I am not a geologist, as far as I can see geological literature use the same convention as in mathematics and physics (e.g. Perry, GCA 31, 1043 (1966); J. Ferguson, Introduction to linear algebra in geology, Chapman&Hall, 1994).

* There is something not quite right with Eqs 7--8: the first element of the left vector is F_{En} , but the corresponding matrix elements are X_{Si}^{Gar} etc. Presumably one of En and Gar should be replaced with the other. Apologies for not spotting this in the initial submission, though it was not as obvious given the formatting issues.

Typo:

Line 178: LHS 2534 currently reads as LH 2534.

Reviewer #3:

Remarks to the Author:

The authors satisfactorily addressed my concerns about the previous draft. I found a few formatting errors and typos which I have listed below. This manuscript is a pleasure to read, and I look forward to seeing it in print.

Line-by-line Comments

Manuscript:

Line 48: Recommend editing to indicate that this is not the standard/recommended method, but how some researchers have done this in the past

Line 181: Stray "wide"?

Supplements:

Equations A1, A3, A5, A7: Perhaps missing a space above as is done with Eq. A2?

Line 542: Perhaps needs a line break before "and"

Line 545: Stray indentations before "(A8)"

Response to Second Round of Reviews:

Reviewer #1 (Remarks to the Author):

I am satisfied with the authors' revisions and now would be happy to see this submission published.

No changes required.

Reviewer #2 (Remarks to the Author):

I thank the authors for either addressing or responding to practically all of my comments. There are some minor issues remaining with the matrix equations in the appendix as well as a new typo, but none of these should not take too long to fix:

* The majority of the matrix indices have been correctly changed to Roman font except for in eqs A1--A2.

We have changed all fonts to Roman since, if we understand R2 correctly, this is proper font for matrix entries. We also now show all equations in R2's preferred format, using column matrix multiplication. To help readers, we also explain that in switching from row to column matrix multiplication, one must take the transpose of the transformation matrix that is otherwise written for row multiplication and reverse the order of the matrices (since matrix multiplication is non-commutative). And to explain this to readers, we re-introduce some of our original row matrix multiplication equations, to show how one is related to the other.

* The matrices and vectors are still transposed with respect to convention. I.e. matrices should be operating on column vectors with each matrix row being multiplied element-wise with the column vector. Although I am not a geologist, as far as I can see geological literature use the same convention as in mathematics and physics (e.g. Perry, GCA 31, 1043 (1966); J. Ferguson, Introduction to linear algebra in geology, Chapman&Hall, 1994).

We have changed the row matrices to column matrices and have made sure that all text within the matrices is in Roman font. As per the Author Checklist, it will now be necessary to view the Supplementary File to view these changes, as they are deleted from the document containing the main text.

* There is something not quite right with Eqs 7--8: the first element of the left vector is F_En, but the corresponding matrix elements are X_Si^Gar etc. Presumably one of En and Gar should be replaced with the other. Apologies for not spotting this in the initial submission, though it was not as obvious given the formatting issues.

Thank you for catching that error. We have changed "En" to "Gar" since we are projecting from garnet (not enstatite).

Typo:

Line 178: LHS 2534 currently reads as LH 2534.

Corrected

Reviewer #3 (Remarks to the Author):

The authors satisfactorily addressed my concerns about the previous draft. I found a few formatting errors and typos which I have listed below. This manuscript is a pleasure to read, and I look forward to seeing it in print.

Line-by-line Comments

Manuscript:

Line 48: Recommend editing to indicate that this is not the standard/recommended method, but how some researchers have done this in the past

We now say “thought by some to provide...”

Line 181: Stray “wide”?

Deleted. Thanks for catching that.

Supplements:

Equations A1, A3, A5, A7: Perhaps missing a space above as is done with Eq. A2?

Line 542: Perhaps needs a line break before “and”

Line 545: Stray indentations before “(A8)”

Line breaks inserted in all these cases, as suggested.